# Grip strength as a mediator in the relationship between physical activity and osteoporosis in older adults: Evidence from two longitudinal cohort studies

Jinguang Gu[1,2©], Bin Zhang[1,2©], Xinyu Long[3], Xiaoqing Wang[1], Weikai Qin[1]*, Yongli Dong[1]*

**1** Wangjing Hospital of China Academy of Chinese Medical Sciences, Beijing, China, **2** China Academy of Chinese Medical Sciences, Beijing, China, **3** Chongqing Rongchang District Traditional Chinese Medicine Hospital, Chongqing, China

☯ These authors contributed equally to this work.
* dongyongli87@163.com (YD); 86uykfh@163.com (WQ)

## Abstract

### Background

Osteoporosis is a skeletal disorder characterized by reduced bone mass and increased fracture risk. With the aging global population, its prevalence is rising, posing a significant public health challenge. Physical activity is considered an effective intervention to reduce osteoporosis risk, but the role of grip strength as a mediator remains underexplored.

### Methods

Data from the English Longitudinal Study of Ageing (ELSA) and the Health and Retirement Study (HRS) were analyzed using generalized linear mixed models (GLMM) and mediation analysis to explore the impact of physical activity on osteoporosis and the role of grip strength. Subgroup analyses accounted for age, gender, and confounding factors.

### Results

The prevalence of osteoporosis was 6.3% in ELSA and 14.1% in HRS. A significant negative correlation was found between physical activity and osteoporosis in both groups (ELSA: OR = 0.234, $P < 0.001$; HRS: OR = 0.638, $P = 0.028$). In those aged ≥65, physical activity had a more pronounced effect (OR = 0.478, $P < 0.001$). Women showed greater benefit. Mediation analysis in the ELSA group revealed that grip strength mediated 28.3% of the effect of physical activity on osteoporosis (ACME = −0.007, P < 0.001).

**Data availability statement:** The data used in this study were the Harmonized ELSA and Harmonized HRS datasets, which are developed and maintained by the Gateway to Global Aging Data. These de-identified datasets are publicly available for research purposes upon registration. Repository Name: Gateway to Global Aging Data Direct URL: https://g2aging.org/harmonized-data/get-data. The authors confirm that they accessed the data via this repository and that other researchers can access the data in the same manner by registering for a free account on the website.

**Funding:** This study was supported by the National Natural Science Foundation of China (grant number 82305278), the Excellent Young Scientific and Technological Talent Cultivation Program of the China Academy of Chinese Medical Sciences (grant number ZZ17-YQ-012), and the Beijing Natural Science Foundation (grant number 7252272). The funders had no role in study design, data collection and analysis, decision to publish, or preparation of the manuscript.

**Competing interests:** All authors state that they have no conflicts of interest regarding the publication of this paper.

**Abbreviations:** ELSA: English Longitudinal Study of Ageing; HRS: Health and Retirement Study; PA: Physical Activity; OP: Osteoporosis; GRIP: Grip Strength; MET: Metabolic Equivalent of Task; BMI: Body Mass Index; ANOVA: Analysis of Variance; GLMM: Generalized Linear Mixed Model; OR: Odds Ratio; CI: Confidence Interval; SD: Standard Deviation; MICE: Multiple Imputation by Chained Equations; ACME: Average CaUSI Mediation Effect; ADE: Average Direct Effect; PM: Proportion Mediated.

## Conclusion

Physical activity, especially resistance training, reduces osteoporosis incidence by enhancing muscle strength, with grip strength playing a mediating role. These findings highlight the importance of physical activity, particularly in older women, for osteoporosis prevention.

---

## 1. Introduction

Osteoporosis is a systemic bone disease characterized by decreased bone mass and degradation of the microstructure of bone tissue, leading to increased bone fragility and susceptibility to fractures [1]. Globally, osteoporosis has become a growing public health problem. It is estimated that more than 200 million people worldwide suffer from osteoporosis [2]. In developed countries such as the UK and the US, the prevalence of osteoporosis and the associated health care costs have continued to increase as the population ages. For example, in the US, there are about 2 million osteoporotic fractures each year [3]. The situation is similar in the UK, where women with osteoporosis suffer from at least one fragility fracture, costing the NHS billions of pounds each year [4]. Therefore, an in-depth understanding of the risk factors for osteoporosis and the exploration of effective prevention and intervention strategies are essential to reduce its social and economic burden.

There are many factors that influence the onset of osteoporosis. In addition to well-recognized contributors such as estrogen deficiency and prolonged immobilization, the protective role of physical activity has become increasingly evident. Cross-sectional and prospective studies have reported that higher levels of physical activity are associated with reduced risks of cardiovascular disease, cancer, and osteoporosis [5,6]. However, evidence from large, multi-regional longitudinal cohort studies specifically addressing osteoporosis remains limited. Physical activity functions to protect bone health primarily by both enhancing mechanical stress and stimulating muscle growth, and studies have shown that resistance activity enhances mechanical stress on bone [7] that increases femoral neck and lumbar spine bone density. In addition, physical activity improves balance and muscle strength and reduces the risk of osteoporotic fractures [8,9]. Exercise stimulates the release of myokines from muscles, which can regulate bone metabolism and enhance bone strength [10,11].

Grip strength reflects the body's overall muscle strength level, and it has been shown that low grip strength is an important factor in osteoporosis risk [12,13], but whether physical activity plays a role in reducing osteoporosis risk by enhancing grip strength has not been reported in the literature. Previous research has predominantly examined the association between physical activity and osteoporosis in a limited scope. While physical activity may impact osteoporosis incidence through various mechanisms, its effect via the enhancement of muscle strength remains underexplored. Therefore, this study aimed to:1) to investigate the relationship between

physical activity levels and osteoporosis in older adults in the UK and US 2) to investigate whether grip strength plays a mediating role in the above relationship.

## 2. Methods

### 2.1. Participants and study design

The data used in this study were derived from two longitudinal cohort studies: the English Longitudinal Study of Ageing (ELSA) and the Health and Retirement Study (HRS). The HRS database began collecting osteoporosis diagnoses from wave 11, so to ensure consistency in the study, we utilized data from waves 4–8 of the ELSA (2008–2016) and waves 11–15 of the HRS (2012–2020). ELSA has been approved by the London Multi-Centre Research Ethics Committee (MREC/01/2/91). The HRS study was approved by the Institutional Review Board of the University of Michigan and the National Institute on Aging (HUM00061128). Informed consent was obtained from all participants in both the English Longitudinal Study of Ageing (ELSA) and the Health and Retirement Study (HRS). For ELSA, participants provided verbal consent for the interview component, which was witnessed and documented by trained interviewers and field staff. Written consent was obtained for other study components, including blood sampling and administrative data linkage. For HRS, participants received a written informed consent information document prior to each interview. At the start of each interview, a confidentiality statement was read to participants, and verbal consent was obtained for participation in the interview.

During the baseline assessment, participants completed evaluations of physical activity, grip strength, and osteoporosis. In addition, data on demographics, lifestyle, physical health, clinical factors, psychological well-being, and economic wealth were also collected. Participants were excluded if they met any of the following criteria: 1) aged under 55; 2) missing data on physical activity, grip strength, or osteoporosis at baseline; 3) no follow-up data completed; 4) no osteoporosis data available during follow-up.

### 2.2. Measurements

**2.2.1. Physical activity.** The physical activity (PA) scoring system proposed in this paper is an extension of Liu et al.'s (2022) methodology [14]. The researchers developed a physical activity (PA) scoring system consisting of three dimensions: First, they assigned a scoring standard for each intensity level (mild, moderate, or vigorous) based on different frequencies (1 = rarely/never, 2 = 1–3 times per month, 3 = weekly or more). Next, each score was weighted according to the average MET value for that intensity level (see S1 Table for details). The overall PA score was obtained by summing the scores from each dimension [15].

The ELSA used three questions to measure participation in mild, moderate, and vigorous physical activity, along with their frequencies. The answer options included: (1) rarely or never, (2) 1–3 times per month, (3) once a week, and (4) more than once a week. The HRS employed three questions related to physical activity, covering vigorous, moderate, or mild activity. The response options for these questions were: (1) daily, (2) several times per week, (3) once a week, (4) 1–3 times per month, and (5) never.

To harmonize the PA scoring between the two cohorts, we mapped the frequency of physical activity responses onto a unified 3-point scale (see S1 Table). For ELSA, responses were coded as follows: "rarely or never" = 1, "1–3 times per month" = 2, and "once a week or more than once a week" = 3. For HRS, responses were coded as: "never" = 1, "1–3 times per month" = 2, and "once a week or several times per week or daily" = 3.The scoring system is the same for all three activity categories.Second, standardized Z-scores are calculated by subtracting the respective mean value and dividing by the baseline standard deviation (SD) of the assigned scores. Third, to capture the differences between mild, moderate, and vigorous PA, the researchers summarized the weighted overall activity Z-scores. The weights were selected based on the estimated metabolic equivalent (MET) of the tasks [16]. After calculation, MET weights of 2.3, 4.4, and 7.2 were assigned

to mild, moderate, and vigorous PA, respectively, which is consistent with previous studies. Detailed information on the weight calculation can be found in S1 Table.

**2.2.2. Grip strength.** In the HRS, grip strength is measured every two years using a Smedley hand dynamometer, while in the ELSA, grip strength is measured every four years. All grip strength tests are performed with the elbow bent at a 90°angle, ensuring that the participant maintains a stable posture in a natural position. For participants who are unable to stand, the wrist is kept in a neutral position during testing, whether standing or sitting. Grip strength is measured for each hand through multiple tests. The final grip strength value is defined as the average of the maximum grip strength values from both hands. This testing method aims to reflect the participants' hand muscle strength, and all measurements are conducted by trained personnel to ensure consistency and accuracy in the testing process [17].

**2.2.3. Osteoporosis.** The diagnosis of osteoporosis in the ELSA and HRS was based on participants' self-reported physician diagnostic information, and this variable was a dichotomous variable (with or without osteoporosis).

**2.2.4. Covariates.** Baseline covariates include sociodemographic factors, behavioral factors, and health-related factors. Specifically, the sociodemographic variables include: age, gender, study cohort (ELSA or HRS), marital status (married or other), household wealth level (low, medium, high), and body mass index (BMI). BMI is calculated using the formula: weight (kg) divided by height (m) squared, with units in $kg/m^2$.

In addition, self-reported health behaviors and health conditions were considered. Health behaviors include smoking status (smoker or non-smoker), with smokers defined as individuals who have smoked in the past or are currently smoking, and drinking status (drinker or non-drinker), with drinkers defined as individuals who have drunk alcohol in the past or are currently drinking. Health-related conditions were based on self-reported physician diagnoses, including hypertension, diabetes, cancer, lung disease, heart disease, stroke, and depression (assessed using the Center for Epidemiologic Studies Depression Scale \[CES-D]). Furthermore, diagnoses of other chronic diseases were also included.

For missing data, this study employed the multiple imputation by chained equations (MICE) method for imputation to improve the accuracy and reliability of data analysis.

## 2.3. Statistical analysis

For descriptive statistics, continuous variables are presented as means±standard deviations (for normally distributed data) or median interquartile ranges (for non-normally distributed data). Categorical variables are presented as frequencies and percentages. Group differences were assessed using analysis of variance (ANOVA), Kruskal-Wallis test (for non-normally distributed data), or chi-square test (for categorical variables).

A generalized linear mixed model (GLMM) was used to calculate odds ratios (ORs) and 95% confidence intervals (95% CIs) to examine the association between physical activity and osteoporosis over an 8-year follow-up period. Potential confounders were considered in the models, and three models were used for adjustment: Model 1: unadjusted; Model 2: adjusted for age, gender, marital status, household wealth level (low, medium, high), and BMI; Model 3: further adjusted for smoking status, alcohol consumption, hypertension, diabetes, stroke, lung disease, heart disease, depression, and other chronic diseases.All covariates included in the models (e.g., BMI, smoking, comorbidities) were treated as time-varying variables to account for changes in participants' status over the follow-up waves.

For key subgroup analyses, fully adjusted models were used to assess the robustness of the association between physical activity and osteoporosis across different age groups (55–64 years vs ≥ 65 years) and gender groups. Sensitivity analysis was performed using binomial logistic regression to examine the association between baseline physical activity and osteoporosis incidence during follow-up, in order to evaluate the robustness of the results.

In this study, a mediation function was used to assess the mediating role of grip strength (grip) in the relationship between physical activity (PA) and osteoporosis (OP) [18]. The mediation analysis was conducted using the R package "mediation." Two models were developed: one assessing the association between the independent variable (physical activity, PA) and the mediator (grip strength, grip) (Path A), and the other assessing the association between the mediator

(grip strength, grip) and the dependent variable (osteoporosis, OP), while controlling for the independent variable (Path B). Path A was estimated using linear regression (since grip strength is a continuous variable), and Path B was estimated using logistic regression (since osteoporosis is a binary variable). Coefficients and 95% CIs were derived to describe the association between physical activity, grip strength, and osteoporosis.

Based on these models, the mediation function was used to estimate the total effect, direct effect, and indirect effect. The bootstrap method was employed to test the significance of the mediation effect, with 1000 resamples to obtain the 95% CIs for the indirect effect and total effect. When the 95% CI did not include zero, the indirect effect was considered statistically significant. The results of this analysis included the average caUSI mediation effect (ACME), average direct effect (ADE), and total effect. The proportion mediated (PM) was calculated as the ratio of ACME to the total effect (ACME/total effect). Additionally, all analyses were adjusted for potential confounders, including age, gender, marital status, wealth, BMI, smoking status, drinking status, hypertension, diabetes, cancer, lung disease, heart disease, stroke, and other chronic diseases.

### 2.4. Ethics

The data used in this study were derived from the English Longitudinal Study of Ageing (ELSA), which has been approved by the ethics committee of University College London. The data from the Health and Retirement Study (HRS) were also used, with approval from the ethics committee of the University of Michigan. All participants signed informed consent forms prior to participation in the study. Participants' privacy and data security were ensured. All data were anonymized and were used solely for scientific research.

## 3. Results

### 3.1. Baseline characteristics of study participants

The participants included individuals aged 55 and above, with 6,862 participants from the ELSA and 6,252 participants from the HRS. The baseline data for ELSA wave 4 consisted of 11,050 participants. After exclusion of the following groups: 1) individuals aged under 55 (n = 1,632), 2) those with missing data on physical activity, grip strength, and osteoporosis at baseline (n = 1,944), 3) those who did not complete the first follow-up (n = 612), and 4) those with missing osteoporosis data during follow-up (n = 0), a total of 6,862 participants were included.

The baseline data for HRS wave 11 consisted of 20,554 participants. After excluding: 1) individuals aged under 55 (n = 2,965), 2) those with missing data on physical activity, grip strength, and osteoporosis at baseline (n = 10,800), 3) those who did not complete the first follow-up (n = 534), and 4) those with missing osteoporosis data during follow-up (n = 3), a total of 6,252 participants were included.

The prevalence of osteoporosis in the ELSA and HRS participants was 6.3% and 14.1%, respectively. The baseline physical activity levels of individuals with osteoporosis were lower than those of the normal group (−0.28 vs. 0.09, −0.04 vs. 0.11), and the baseline grip strength of osteoporosis patients was also lower than that of the normal group (21.07 vs. 30.04, 21.86 vs. 29.95). Furthermore, the prevalence of osteoporosis was significantly associated with age, heart disease, and BMI levels compared to the normal group. Specific details are presented in Table 1.

### 3.2. Association between physical activity and osteoporosis

The study used a generalized linear mixed model (GLMM) to examine the association between physical activity and osteoporosis over an 8-year follow-up period, adjusting for relevant variables. The results showed that in the ELSA cohort, increased physical activity was associated with a 76.6% reduction in the risk of osteoporosis, with an OR of 0.234 (95% CI: 0.161, 0.341, $P<0.001$). Similar results were observed in the HRS cohort, where increased physical activity was associated with a 36.2% reduction in the risk of osteoporosis, with an OR of 0.638 (95% CI: 0.428, 0.953, $P=0.0281$). The

**Table 1. Baseline characteristics of study participants.**

| Variables | ELSA(n=6862) | | | | HRS(n=6252) | | | |
|---|---|---|---|---|---|---|---|---|
| | Overall | No Osteoporosis | Osteoporosis | p | Overall | No Osteoporosis | Osteoporosis | p |
| n | 6862 | 6429 | 433 | | 6252 | 5372 | 880 | |
| PA:mean (SD) | 0.06 (0.75) | 0.09 (0.74) | -0.28 (0.79) | <0.001 | 0.09 (0.77) | 0.11 (0.77) | -0.04 (0.75) | <0.001 |
| Handgrip (kg):mean (SD) | 29.48 (10.96) | 30.04 (10.87) | 21.07 (8.53) | <0.001 | 28.81 (10.52) | 29.95 (10.51) | 21.86 (7.44) | <0.001 |
| Age(years): Mean (SD) | 67.01 (8.51) | 66.74 (8.40) | 71.01 (9.13) | <0.001 | 68.32 (9.56) | 67.84 (9.50) | 71.20 (9.48) | <0.001 |
| Married(Yes) | 4629 (67.5) | 4411 (68.6) | 218 (50.3) | <0.001 | 3627 (58.0) | 3195 (59.5) | 432 (49.1) | <0.001 |
| Gender | | | | | | | | |
| Male | 3119 (45.5) | 3066 (47.7) | 53 (12.2) | <0.001 | 2655 (42.5) | 2566 (47.8) | 89 (10.1) | <0.001 |
| Female | 3743 (54.5) | 3363 (52.3) | 380 (87.8) | | 3597 (57.5) | 2806 (52.2) | 791 (89.9) | |
| Wealth | | | | | | | | |
| Low | 2266 (33.0) | 2076 (32.3) | 190 (43.9) | <0.001 | 2002 (32.0) | 1733 (32.3) | 269 (30.6) | 0.546 |
| Mid | 2505 (36.5) | 2361 (36.7) | 144 (33.3) | | 2231 (35.7) | 1905 (35.5) | 326 (37.0) | |
| High | 2091 (30.5) | 1992 (31.0) | 99 (22.9) | | 2019 (32.3) | 1734 (32.3) | 285 (32.4) | |
| Bmi(kg/m²): Mean (SD) | 28.39 (5.20) | 28.46 (5.18) | 27.42 (5.36) | <0.001 | 29.70 (6.13) | 29.93 (6.09) | 28.26 (6.21) | <0.001 |
| Smoke(Yes) | 860 (12.5) | 792 (12.3) | 68 (15.7) | 0.048 | 812 (13.1) | 705 (13.2) | 107 (12.2) | <0.001 |
| Drink(Yes) | 5529 (89.5) | 5230 (90.2) | 299 (77.7) | <0.001 | 3513 (56.2) | 3104 (57.8) | 409 (46.5) | 0.465 |
| Hypertension(Yes) | 2820 (41.1) | 2619 (40.7) | 201 (46.4) | 0.023 | 3814 (61.0) | 3280 (61.1) | 534 (60.7) | <0.001 |
| Diabetes(Yes) | 625 (9.1) | 584 (9.1) | 41 (9.5) | 0.855 | 1478 (23.6) | 1309 (24.4) | 169 (19.2) | 0.862 |
| Cancer(Yes) | 584 (8.5) | 528 (8.2) | 56 (12.9) | 0.001 | 917 (14.7) | 769 (14.3) | 148 (16.8) | 0.001 |
| Pulmonary Disease(Yes) | 389 (5.7) | 335 (5.2) | 54 (12.5) | <0.001 | 573 (9.2) | 436 (8.1) | 137 (15.6) | 0.058 |
| Heart Disease(Yes) | 1137 (16.6) | 1021 (15.9) | 116 (26.8) | <0.001 | 1479 (23.7) | 1236 (23.0) | 243 (27.6) | <0.001 |
| Stroke(Yes) | 260 (3.8) | 235 (3.7) | 25 (5.8) | 0.035 | 482 (7.7) | 403 (7.5) | 79 (9.0) | 0.003 |
| Other Chronic Diseases(Yes) | 3761 (54.8) | 3464 (53.9) | 297 (68.6) | <0.001 | 1719 (27.5) | 1391 (25.9) | 328 (37.3) | 0.146 |

Data are presented as N (%) for categorical variables or Mean (SD) for continuous variables. P values were derived from Chi-square tests for categorical variables and independent t-tests for continuous variables, comparing the 'No Osteoporosis' vs. 'Osteoporosis' groups.

1 PA (physical activity) is presented as the weighted overall activity Z-score; unit: standard deviation (SD).

2 Handgrip (kg): Handgrip strength measured in kilograms.

Abbreviations: BMI, body mass index; ELSA, English Longitudinal Study of Ageing; HRS, Health and Retirement Study; PA, physical activity; SD, standard deviation.

model also revealed the significance of other variables, including age, diabetes, asthma, and mental health, which had a significant impact on the occurrence of osteoporosis. These details are presented in Table 2.

### 3.3. Subgroups and sensitivity analysis

Subgroup analyses by age and gender were conducted. The results showed that in the ELSA cohort, the effect of physical activity on osteoporosis was more pronounced in the ≥65 years group (*P*<0.001), with a nearly 50% reduction in the risk of osteoporosis (OR = 0.478, 95% CI: 0.346, 0.661). This indicates that the protective effect of physical activity on osteoporosis becomes more significant with age. Although physical activity had a significant protective effect against osteoporosis in the ELSA cohort, the results in the HRS cohort did not reach statistical significance, possibly due to sample differences, data processing methods, or cultural differences. These findings are shown in Table 3.

For gender analysis, the effect in the ELSA male group was not significant, while the effect in the female group was significant (OR = 0.697, 95% CI: 0.492, 0.986, *P*=0.042). In the HRS cohort, the female group showed a moderate significance (OR = 0.852, 95% CI: 0.747, 0.973, *P*=0.018). This suggests that physical activity has a more pronounced protective effect against osteoporosis in Women. However, the interaction between gender and physical activity was not

**Table 2. Association between physical activity and osteoporosis over time.**

| Variables | ELSA | | HRS | |
|---|---|---|---|---|
| | OR (95% CI) | P value | OR (95% CI) | P value |
| MODEL 1 | | | | |
| Intercept term | 0.000 (0.000, 0.00) | <0.001 | 0.000 (0.000, 0.000) | <0.001 |
| PA (Z-score) | 0.259 (0.203, 0.33) | <0.001 | 0.443 (0.379, 0.518) | <0.001 |
| Standard deviation of intercept | 17.620 | NA | 15.780 | NA |
| MODEL 2 | | | | |
| Intercept term | 0.000 (0.000, 0.000) | <0.001 | 0.000 (0.000, 0.000) | <0.001 |
| PA (Z-score) | 0.256 (0.148, 0.443) | <0.001 | 0.774 (0.619, 0.969) | 0.025 |
| Age (years) | 1.670 (1.373, 2.030) | <0.001 | 1.935 (1.813, 2.066) | <0.001 |
| Gender (Male)† | 0.488 (0.166, 1.435) | 0.192 | 0.001 (0.000, 0.004) | <0.001 |
| Marital Status (Married) | 1.087 (0.467, 2.532) | 0.847 | 1.172 (0.721, 1.904) | 0.523 |
| Wealth | 0.532 (0.293, 0.968) | 0.039 | 1.152 (0.852, 1.556) | 0.358 |
| BMI (kg/m²) | 0.702 (0.599, 0.822) | <0.001 | 0.989 (0.968, 1.010) | 0.306 |
| Standard deviation of intercept | 16.928 | NA | 26.203 | NA |
| MODEL 3 | | | | |
| Intercept term | 0.000 (0.000, 0.000) | <0.001 | 0.000 (0.000, 0.000) | <0.001 |
| PA (Z-score) | 0.234 (0.161, 0.341) | <0.001 | 0.638 (0.428, 0.953) | 0.028 |
| Age (years) | 1.202 (1.148, 1.258) | <0.001 | 1.295 (1.233, 1.359) | <0.001 |
| Gender (Male)† | 0.106 (0.049, 0.231) | <0.001 | 0.001 (0.000, 0.003) | <0.001 |
| Marital Status (Married) | 1.175 (0.642, 2.151) | 0.600 | 2.751 (1.252, 6.040) | 0.012 |
| Wealth | 1.138 (0.808, 1.603) | 0.460 | 0.662 (0.413, 1.060) | 0.086 |
| BMI (kg/m²) | 0.812 (0.757, 0.870) | <0.001 | 0.985 (0.947, 1.025) | 0.461 |
| Smoking Status (Yes) | 0.185 (0.055, 0.616) | 0.006 | 0.955 (0.263, 3.466) | 0.944 |
| Drinking Status (Yes) | 0.308 (0.164, 0.578) | <0.001 | 0.936 (0.493, 1.776) | 0.840 |
| Hypertension (Yes) | 0.340 (0.186, 0.622) | <0.001 | 0.237 (0.100, 0.565) | 0.001 |
| Diabetes (Yes) | 0.200 (0.066, 0.604) | 0.004 | 0.275 (0.107, 0.705) | 0.007 |
| Cancer (Yes) | 2.152 (1.087, 4.258) | 0.028 | 1.205 (0.454, 3.200) | 0.708 |
| Lung Disease (Yes) | 0.901 (0.350, 2.320) | 0.828 | 12.609 (4.087, 38.903) | <0.001 |
| Heart Disease (Yes) | 1.087 (0.600, 1.969) | 0.783 | 0.421 (0.195, 0.910) | 0.028 |
| Stroke (Yes) | 4.361 (1.679, 11.332) | 0.003 | 1.046 (0.328, 3.338) | 0.940 |
| Psychological Health Issues (Yes) | 2.591 (1.031, 6.513) | 0.043 | 18.849 (7.402, 47.995) | <0.001 |
| Asthma (Yes) | 1.880 (0.855, 4.131) | 0.116 | 2.871 (0.416, 19.838) | 0.285 |
| High Cholesterol (Yes) | 1.180 (0.694, 2.008) | 0.541 | 3.003 (1.366, 6.601) | 0.006 |
| Standard deviation of intercept | 11.175 | NA | 9.808 | NA |

Notes: Values are presented as OR (95% CI). All P values are two-sided.

Model 1: Unadjusted.

Model 2: Adjusted for Age, Gender, Marital Status, Wealth, and BMI.

Model 3: Adjusted for Model 2 variables plus Smoking Status, Drinking Status, Hypertension, Diabetes, Cancer, Lung Disease, Heart Disease, Stroke, Psychological Health Issues, Asthma, and High Cholesterol.

Covariate Definitions and Units:

1 PA (Physical Activity): OR represents risk change per 1-Standard Deviation (1-SD) increase in Z-score.

2 BMI: OR represents risk change per 1 kg/m² increase.

3 Age: OR represents risk change per 1 year increase.

4 Categorical Variables (Reference Groups):

†Gender: Reference group is Female. OR < 1 indicates Males have lower risk.

Marital Status: Reference group is Unmarried.

*(Continued)*

**Table 2.** (Continued)

Wealth: Reference group is Low.

Binary conditions (e.g., Smoking, Diseases): Reference group is No (absence of condition).

5 Time-Varying Covariates: All covariates were included as time-varying variables to capture changes during the follow-up.

Abbreviations: BMI, body mass index; CI, confidence interval; ELSA, English Longitudinal Study of Aging; GLMM, generalized linear mixed model; HRS, Health and Retirement Study; OR, odds ratio; PA, physical activity.

**Table 3. Subgroup analysis of the relationship between physical activity and osteoporosis (age and gender).**

| cVariables | ELSA | | | HRS | | |
|---|---|---|---|---|---|---|
| | OR (95% CI) | P value | P for interaction | OR (95% CI) | P value | P for interaction |
| Age | | | | | | |
| 55-64 | 0.209 (0.041, 1.052) | 0.058 | <0.001 | 0.937 (0.731, 1.200) | 0.607 | 0.355 |
| ≥65 | 0.478 (0.346, 0.661) | <0.001 | | 0.893 (0.780, 1.023) | 0.103 | |
| Gender | | | | | | |
| Male | 0.723 (0.428, 1.223) | 0.226 | 0.857 | 1.145 (0.864, 1.518) | 0.347 | 0.547 |
| Female | 0.697 (0.492, 0.986) | 0.042 | | 0.852 (0.747, 0.973) | 0.018 | |

significant (ELSA: $P=0.857$, HRS: $P=0.547$), indicating that the gender difference is not very pronounced. These results are presented in Table 3.

Sensitivity analysis was performed using a binomial logistic regression (logit) model to calculate odds ratios (ORs) and 95% confidence intervals (95% CIs) to examine the association between baseline physical activity and osteoporosis incidence during follow-up. The aim was to assess whether the association between physical activity and osteoporosis remained robust. The results showed that after applying the binomial logistic regression model, physical activity remained negatively associated with osteoporosis. After controlling for multiple confounders (Model 3), the OR for ELSA was 0.741 (95% CI: 0.659, 0.833, $P<0.001$) and for HRS, the OR was 0.907 (95% CI: 0.829, 0.992, $P<0.033$), both of which were significant, further supporting the main findings of the study. These results are presented in Table 4.

### 3.4. Mediating effects of grip strength

The analysis results showed that after adjusting for covariates such as age, gender, and marital status, the total effect of physical activity (PA) on osteoporosis in the ELSA cohort was −0.025 (95% CI: −0.034, −0.016). The indirect effect of PA on osteoporosis through grip strength (ACME) was −0.007 (95% CI: −0.009, −0.004), indicating that PA can reduce the risk of osteoporosis by improving grip strength. The direct effect of PA on osteoporosis (ADE) was −0.018 (95% CI: −0.027, −0.008). The mediation proportion analysis showed that grip strength explained 28.3% of the total effect of PA on osteoporosis (95% CI: 0.175, 0.504).

A similar result was found in the HRS cohort. The indirect effect (ACME) was −0.007 (95% CI: −0.009, −0.004), suggesting that grip strength still plays a marginally significant mediating role. The total effect was −0.012 (95% CI: −0.025, 0.001), and the direct effect was −0.006 (95% CI: −0.020, 0.008). Grip strength explained 54.1% of the total effect of PA on osteoporosis (95% CI: 0.000, 3.098), though the total effect in HRS was marginally significant ($P=0.068$). The detailed results can be found in Table 5, Figs 1 and 2.

### 4. Discussion

This study is the first to use a multinational database to explore the relationship between physical activity and the incidence of osteoporosis, revealing that grip strength plays a mediating role in this association. The results showed a significant negative correlation between physical activity and osteoporosis in both countries. In the mediation analysis, grip

**Table 4. Sensitivity analysis: Association between baseline physical activity and osteoporosis incidence during follow-up.**

| Variables | ELSA | | HRS | |
|---|---|---|---|---|
| | OR (95% CI) | P value | OR (95% CI) | P value |
| MODEL1 | | | | |
| Physical Activity | 0.591 (0.535,0.653) | <0.001 | 0.801 (0.743,0.864) | <0.001 |
| MODEL2 | | | | |
| Physical Activity | 0.687 (0.614,0.770) | <0.001 | 0.877 (0.804,0.956) | <0.003 |
| MODEL3 | | | | |
| Physical Activity | 0.741 (0.659,0.833) | <0.001 | 0.907 (0.829,0.992) | <0.033 |

Model 1: Unadjusted.

Model 2: Adjusted for Age, Gender, Marital Status, Wealth, BMI.

Model 3: Model 2 + adjusted for Smoking Status, Drinking Status, Hypertension, Diabetes, Cancer, Lung Disease, Heart Disease, Stroke, Psychological Health Issues, Asthma, High Cholesterol.

**Table 5. Mediating effect of grip strength between physical activity and osteoporosis risk.**

| Effect Type | Elsa | | Hrs | |
|---|---|---|---|---|
| | Effect Value (95% CI) | P-value | Effect Value (95% CI) | P-value |
| ACME (Indirect Effect) | −0.007 (−0.009, −0.004) | <0.001 | −0.007 (−0.009, −0.004) | <0.001 |
| ADE (Direct Effect) | −0.018 (−0.027, −0.008) | <0.001 | −0.006 (−0.020, 0.008) | 0.410 |
| Total Effect | −0.025 (−0.034, −0.016) | <0.001 | −0.012 (−0.025, 0.001) | 0.068 |
| Mediation Proportion | 0.283 (0.175, 0.504) | <0.001 | 0.541 (0.000, 3.098) | 0.068 |

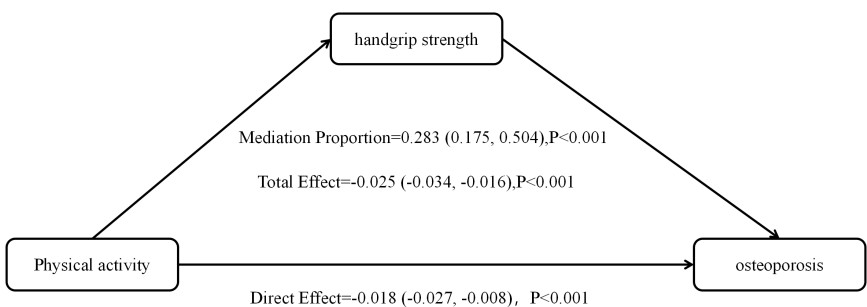

**Fig 1. Effect plot of mediation analysis for ELSA group.**

strength partially mediated the relationship between physical activity and osteoporosis, with the mediation effect exceeding 20% in both cohorts.

This study is consistent with previous research, which indicates that physical activity can reduce the risk of osteoporosis. A study based on the Korean population found that high-intensity physical activity is associated with a lower prevalence of osteopenia and osteoporosis, as well as improved bone mineral density and vitamin D levels [19]. Additionally, physical activity has been shown to reduce the incidence of osteoporotic fractures [20]. However, the previous studies were cross-sectional and conducted in a single country. This study, based on the English Longitudinal Study of Ageing (ELSA) and the Health and Retirement Study (HRS) in the US, spans an 8-year period, strengthening both the population

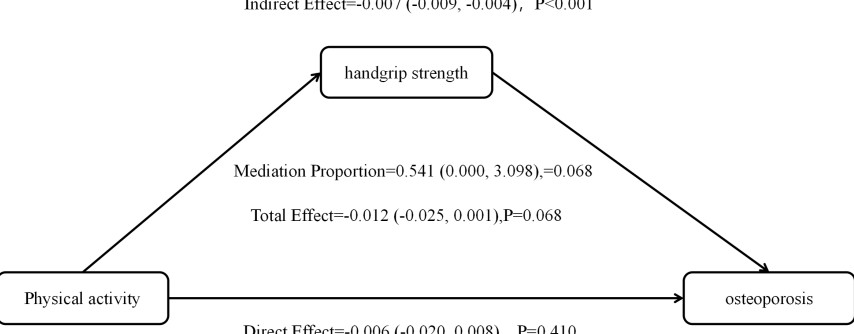

**Fig 2. Plot of mediating effect analysis in HRS group.**

and the time frame, which enhances its credibility. The results show that individuals who engage in regular physical activity, have good mental health, and do not have asthma are at a lower risk of osteoporosis. In the UK, the risk of osteoporosis decreased by about 76.6% in those with increased physical activity, while in the US, this risk decreased by 36.2%. This suggests that in the UK, the effect of physical activity on osteoporosis may be more pronounced. We hypothesize that this phenomenon may be related to differences in the healthcare systems. The UK's National Health Service (NHS) provides free or low-cost healthcare services to all residents, including routine health checks, disease treatment, and rehabilitation for older adults. In contrast, the US healthcare system is primarily based on private insurance, and although the medical standard is high, the health screening rate for low-income groups is lower [21]. Older people in the UK can find out about their bone density levels through free health screenings and take action accordingly.

Subsequently, we conducted subgroup analyses by gender and age. The results showed that the interaction effect was significant only in the ELSA cohort, where individuals aged 65 and above benefited more from physical activity. This suggests that it is never too late to start exercising. When analyzed separately, the female group showed greater benefits from physical activity in terms of bone protection. This is likely related to the higher incidence of osteoporosis in postmenopaUSI women. Overall, our study highlights the important role of age and gender in the impact of physical activity on osteoporosis. Specifically, for older adults and women, early and consistent physical activity can significantly reduce the risk of developing osteoporosis [22,23].

Physical activity plays an important role in protecting bone health through two key mechanisms: enhancing mechanical stress and stimulating muscle growth. Regular exercise continuously stimulates an increase in bone diameter throughout the lifespan. This bone diameter increase, induced by exercise, counteracts bone thinning and the increase in bone porosity through mechanical loading, thereby reducing the risk of fractures [24]. Exercise-mediated mechanical loading enhances bone mineral density for the treatment of osteoporosis in suspension-tailed mice [25]. Research has shown that resistance training enhances the mechanical loading of the skeleton, which in turn helps to increase bone density in the femoral neck and lumbar spine [7]. In addition, physical activity improves balance and muscle strength and reduces the risk of osteoporotic fractures [8,9].

Research confirms that resistance training significantly improves muscle strength (e.g., grip strength), which is a key source of mechanical loading of the skeleton [26]. A 12-week resistance exercise program significantly improved muscle strength and balance in postmenopaUSI women with osteoporosis, leading to comprehensive improvements in functional outcomes [27]. Additionally, individuals with low muscle mass have a significantly higher risk of osteoporosis (men HR: 1.30 [95% CI: 1.03 to 1.63], women HR: 1.66 [95% CI: 1.33 to 2.08]) [28]. Clinical studies have also shown that older men and women with muscle impairment (i.e., low muscle mass, low muscle strength, and low physical function) have lower aBMD values [29]. These findings indirectly confirm the protective role of muscle strength in bone health.

Skeletal muscle contraction can partially reverse the incidence of osteoporosis. Skeletal muscle promotes osteoblast differentiation by releasing exosomes (DN+ES-Exo) [30]. Myokines such as Irisin secreted by muscles can enhance osteogenesis, while inflammatory factors associated with age-related sarcopenia (e.g., IL-6) accelerate bone loss [31]. Researchers have found that the conditioned medium of skeletal muscle (Mu-EV) can promote osteogenic differentiation of BMSCs and inhibit osteoclast formation from monocytes, thus affecting the balance of bone remodeling. Additionally, exogenous Mu-EV from normal skeletal muscle has been shown to reverse disuse-induced osteoporosis. Grip strength, which reflects overall muscle strength, has been identified as an important factor for osteoporosis risk, with low grip strength being a significant risk factor for osteoporosis [12,13]. Notably, grip strength is not merely an isolated metric of muscle strength. According to the latest diagnostic consensus from European (EWGSOP2), Asian (AWGS), and other international bodies, low grip strength is a core diagnostic criterion and the primary screening indicator for Sarcopenia [32,33]. In geriatric medicine, Sarcopenia and Osteoporosis are widely regarded as the "Hazardous Duo," as these conditions frequently co-exist and synergistically increase the risk of falls, fractures, and mortality [34]. The two conditions are closely linked pathophysiologically.Therefore, the use of grip strength as a mediating variable in our study can be interpreted, from a clinical perspective, as an exploration of the mediating role of Sarcopenia in the relationship between physical activity and osteoporosis. Our mediation analysis substantiates this, revealing that the indirect effect of PA mediated through grip strength accounted for 28.3% of the total effect in the ELSA cohort. This finding strongly suggests that a significant portion of the protective effect of physical activity (PA) on bone health is achieved by maintaining or enhancing muscle function (i.e., preventing or ameliorating sarcopenia).

This study has practical implications: Firstly, community health organizations should regularly assess grip strength in older adults and provide bone mineral density screening and calcium supplementation for those with low grip strength. Secondly, policymakers should focus on educational activities, emphasizing the benefits of resistance training and physical activities that enhance muscle strength for bone health. Finally, the government could invest in public fitness equipment, with a greater emphasis on resistance muscle training.

This study has several limitations: (1) Potential bias in physical activity measurement: Physical activity levels were collected through self-reported questionnaires, which may be subject to recall bias or subjectivity. This could lead to either an underestimation or overestimation of the association between physical activity and osteoporosis. (2) Limitations of grip strength as a single indicator: Grip strength only reflects upper limb muscle strength, which may not fully represent overall muscle function or the contribution of other muscle groups (such as lower limbs) related to osteoporosis. This limits the comprehensive assessment of the mediating effect. (3) Insufficient adjustment for confounding factors: Although the study adjusted for confounding factors such as age, gender, marital status, household wealth, and BMI, other potential influencing factors (such as nutritional intake, vitamin D levels, hormone replacement therapy, or genetic factors) may not have been fully incorporated, potentially affecting the robustness of the results. (4) Lack of biomarker data, such as bone mineral density: This study did not include biomarker data like bone mineral density, which is crucial for assessing the risk of osteoporosis. Future research could validate the effects of physical activity and grip strength on osteoporosis using more direct biological markers.

## 5. Conclusion

This study utilized two elderly cohort studies: the English Longitudinal Study of Ageing (ELSA) and the Health and Retirement Study (HRS), systematically assessing the association between the physical activity, grip strength, mental health, and osteoporosis. The study found that regular physical activity is associated with a lower incidence of osteoporosis, with grip strength as a partial mediator. Further mediation analysis revealed that grip strength partially mediated the relationship between physical activity and osteoporosis. Subgroup analysis indicated that the impact of physical activity was particularly significant in the elderly population, especially in those aged ≥65 years. The study highlights that physical activity, through its effect on muscle strength, reduces the incidence of osteoporosis and emphasizes the importance of resistance

training and other physical activities for bone health protection, particularly in older women. While the findings suggest a protective role of physical activity, further research is needed to confirm caUSlity due to the observational nature of the study.

The findings of this study provide clear guidance for clinicians in the prevention and treatment of osteoporosis. The following specific recommendations are proposed:

(1) Personalized Exercise Prescriptions: Clinicians should recommend tailored resistance training programs based on patients' physical condition, age, and gender. For instance, 2–3 sessions per week of moderate-intensity resistance training (e.g., using dumbbells, resistance bands, or bodyweight exercises) lasting 30–60 minutes can significantly improve bone health. For older women, combining this with low-impact aerobic activities (such as brisk walking or swimming) can further enhance overall health outcomes.

(2) Comprehensive Assessment and Monitoring: Clinicians are advised to regularly monitor patients' bone health and exercise outcomes using tools such as grip strength tests, bone mineral density assessments (e.g., Dual-Energy X-ray Absorptiometry, DXA), and other functional evaluations to optimize intervention strategies.

(3) Multidisciplinary Collaboration: Effective management of osteoporosis requires a collaborative approach involving orthopedic specialists, physical therapists, and nutritionists. Together, they can develop comprehensive intervention plans that integrate exercise, nutrition (e.g., calcium and vitamin D supplementation), and lifestyle modifications.

These clinical recommendations can help older adults, particularly women, reduce the risk of osteoporosis-related fractures, improve quality of life, and decrease long-term healthcare needs.

Public Health Policy Implications:Osteoporosis and its associated fractures place a growing social and economic burden, particularly in aging societies. The study's findings offer the following implications for public health policy:

(1) Promotion of Community-Based Exercise Programs: Governments and community organizations should develop low-cost or free exercise programs tailored for older adults, such as group resistance training classes, tai chi, or adapted fitness programs. These initiatives can be delivered through community centers, retirement homes, or online platforms to enhance accessibility and participation.

(2) Health Education and Awareness Campaigns: Public media, community workshops, and health promotion materials should be used to raise awareness about the importance of physical activity for bone health. Efforts should focus on addressing concerns about exercise safety among older adults, emphasizing the safety and benefits of moderate resistance training.

By implementing these measures, public health policies can effectively reduce the incidence of osteoporosis-related fractures at the community level, alleviating the burden on healthcare systems and society. Although this study is based on Western populations, its findings hold potential relevance for Asian populations. Future localized research is needed to refine intervention strategies. Through integrated clinical and public health efforts, the prevention and management of osteoporosis can significantly improve health outcomes for older adults.

## Supporting information

**S1 Table. Physical activity intensity categories, activity types, MET values, codes, and MET weights.** a Activity type was selected based on examples showed to participants when investigating frequency of physical activity. b MET: metabolic equivalent of tasks. MET estimates were derived according to 2011 Compendium of Physical Activities. c Code represented exact type of activities, for activity corresponding to multiple potential types, Li et al. (2022) used mean of MET values from these activities.
(DOCX)

## Acknowledgments

We sincerely thank the participants and staff of the English Longitudinal Study of Ageing (ELSA) and the Health and Retirement Study (HRS) for providing access to their high-quality data, which made this research possible.

## Author contributions

**Conceptualization:** Jinguang Gu.

**Methodology:** Bin Zhang.

**Software:** Xinyu Long, Xiaoqing Wang.

**Supervision:** Weikai Qin.

**Visualization:** Bin Zhang, Yongli Dong.

**Writing – original draft:** Jinguang Gu, Bin Zhang.

**Writing – review & editing:** Jinguang Gu, Bin Zhang.

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
