## [Decision Letter · Decision Letter 0]

12 Nov 2025

Dear Dr. Dong,

Thank you for submitting your manuscript to PLOS ONE. After careful consideration, we feel that it has merit but does not fully meet PLOS ONE’s publication criteria as it currently stands. Therefore, we invite you to submit a revised version of the manuscript that addresses the points raised during the review process.

The manuscript has been reviewed by 2 experts in the field.  While the findings are interesting, there remain severe deficiencies with the manuscript. The author is invited to revise and resubmit the manuscript. The authors should respond to each of the comments.

We look forward to receiving your revised manuscript.

Kind regards,

Dengshun Miao

Academic Editor

PLOS ONE

[This study was supported by the National Natural Science Foundation of China (grant number 82305278), the Excellent Young Scientific and Technological Talent Cultivation Program of the China Academy of Chinese Medical Sciences (grant number ZZ17-YQ-012), and the Beijing Natural Science Foundation (grant number 7252272).].

Additional Editor Comments (if provided):

Reviewers' comments:

Reviewer's Responses to Questions

**Comments to the Author**

1. Is the manuscript technically sound, and do the data support the conclusions?

Reviewer #1: Partly

Reviewer #2: Yes

2. Has the statistical analysis been performed appropriately and rigorously?

Reviewer #1: I Don't Know

Reviewer #2: Yes

3. Have the authors made all data underlying the findings in their manuscript fully available?

Reviewer #1: Yes

Reviewer #2: Yes

4. Is the manuscript presented in an intelligible fashion and written in standard English?

Reviewer #1: Yes

Reviewer #2: Yes

Reviewer #1: In this study, they investigated the contribution of grip strength in the relationship between physical activity and osteoporosis incidence in older adults based on two large cohort studies (ELSA, GLMM) from UK and US.

The data is useful and the manuscript is well written.

There are several issues, which should be addressed.

1. Since BMI is considered in most standard studies and BMI might influence grip strength and physical activity, the analyses of this study should be adjusted with BMI. Just limitation description is not enough.

2. Grip strength is included in many diagnosis criteria for sarcopenia and/or osteosarcopenia. The results of this study should be discussed as for the association with sarcopenia.

3. Several minor errors should be carefully corrected. For example, there are double periods in several sentences.

Reviewer #2: This is a well-written manuscript. And I am happy to review this work.

The authors have done well especially considering the large cohorts used. This has added a strong weight to the work.

The authors investigate whether grip strength mediates the association between physical activity (PA) and osteoporosis (OP) using longitudinal data from ELSA (waves 4–8; 2008–2016) and HRS (waves 11–15; 2012–2020). The authors employ generalised linear mixed models (GLMMs) to model operational risk, conduct subgroup and sensitivity analyses, and execute mediation analysis using R mediation. The study identifies an inverse association between PA and overall performance in both cohorts. However the following observations were made for consideration:

1. The PA scoring is extended from prior work, but ELSA and HRS questionnaires and response scales differ. I suggest a more detail on harmonization.

2. Authors to clarify whether covariates (e.g., smoking, comorbidities, BMI) were treated as time-varying and how missing data across waves was handled

3. The age subgroup categorises uses “<65 vs ≥65,” yet the eligibility criteria exclude those under 55. Please clarify the age intervals, as the text occasionally suggests a range of 50–64.

4. P-values (e.g., 0.02813) display too many decimal figures; please standardize to 2–3 decimals.

5. Language & formatting: Numerous typographical and grammatical issues were identified (e.g., “Famale”, spacing, punctuation, inconsistent use of decimals/commas). I recommend extra professional English editing.

6. Tables/Figures:

i) Ensure all tables have clear titles, and consistent decimal places.

ii) In Table 2, ensure all covariates ae defined identically across cohorts; verify units/scales for BMI and PA z-scores

iii) Citations: Check that all in-text bracketed references are present in the reference list and conform to journal style; some appear duplicated or partially truncated in the draft.

7. Numbers in text: Match all prevalence, ORs, CIs, and mediation percentages between Abstract/Results/Tables. Some inconsistence exist

8. Page numbering: Ensure pages are numbered for easy identifications and review.

**Do you want your identity to be public for this peer review?** For information about this choice, including consent withdrawal, please see our Privacy Policy

Reviewer #1: No

Reviewer #2: No

---

## [Author Response · Author response to Decision Letter 1]

24 Nov 2025

Response to Reviewers (PONE-D-25-47126)

Manuscript Title: Grip Strength as a Mediator in the Relationship Between Physical Activity and Osteoporosis in Older Adults: Evidence from Two Longitudinal Cohort Studies

Dear Dr Dengshun Miao (Academic Editor) and Reviewers：

We are grateful for your valuable comments and constructive suggestions regarding our manuscript (PONE-D-25-47126). We have carefully considered all comments and believe these suggestions have substantially enhanced the quality of our research.

The manuscript has been comprehensively revised in accordance with all your recommendations. (Revisions are highlighted in blue.) We have endeavoured to address all issues and believe the revised manuscript demonstrates significant improvement in both scientific rigour and clarity.

Below is our point-by-point response to each comment raised by the Editor and Reviewers.

1.Response to Academic Editor and Journal Requirements

Point Raised Our Response

Journal Requirements 1: Formatting Requirements

(Please ensure that your manuscript meets PLOS ONE's style requirements, including those for file naming...) We have carefully reviewed the formatting requirements for PLOS ONE, including the main text, title page, author information, and file naming conventions, and have amended the manuscript accordingly.

Journal Requirement 2: Statement on the Role of Funders

(Please state what role the funders took in the study... "The funders had no role...") We acknowledge the role of the funder in the research design, publication decisions, or manuscript preparation. We have updated this statement in our cover letter.

Journal Requirement 3: Data Availability Statement

(Please address the following prompts regarding data restrictions...) Thank you for the opportunity to clarify our Data Availability Statement. We confirm the following:

a) There are ethical and legal restrictions on sharing the dataset as this study utilizes third-party data. We, the authors, do not have the legal right to publicly share, republish, or redistribute these datasets.

* The ELSA (English Longitudinal Study of Ageing) data contains potentially identifying and sensitive participant information. These data are managed by the UK Data Service on behalf of the original data creators (NatCen Social Research, University College London, Institute for Fiscal Studies, and the University of Manchester). Access is restricted by the usage agreements imposed by these institutions.

* The HRS (Health and Retirement Study) data also contains sensitive participant information and is sponsored and managed by the University of Michigan. Data distribution is restricted by a formal registration and data use agreement to protect participant confidentiality.

b) Consequently, we are unable to upload these data to a public repository or provide them as Supporting Information files.

c) Other qualified researchers can access these data by applying directly to the original data providers. We have updated our Data Availability Statement in the manuscript and submission form to provide precise access information:

* ELSA data are available from the UK Data Service at: https://www.ukdataservice.ac.uk/ (or https://www.elsa-project.ac.uk/). Data requests must be submitted via their registration and application process.

* HRS data are available from the University of Michigan at: https://hrs.isr.umich.edu/data-products. Researchers must register and agree to the data use terms on their website to download the data.

Journal Requirement 4: Corresponding author ORCID iD

(PLOS requires an ORCID iD for the corresponding author...) The corresponding author (Dong Yongli) has confirmed that their ORCID iD (0009-0003-7847-1398) has been successfully verified within the Editorial Manager system.

21(Response to Reviewer #1)

Point Raised Our Response

Comment 1.1: BMI Calibration

(Since BMI... should be adjusted with BMI. Just limitation description is not enough.) We thank the reviewer for raising this important point. We fully agree that BMI is a key confounding variable that may influence physical activity, grip strength, and osteoporosis simultaneously.

We would like to clarify that BMI was indeed included as a core covariate in our analysis in the original submission.

As shown in Table 2 of our manuscript, BMI has already been fully adjusted for in Model 2 and the final Model 3.In the ELSA cohort, the result for BMI in Model 3 was OR = 0.812, 95% CI: 0.757–0.870, P < 0.001.

Comment 1.2: Sarcopenia Discussion

(Grip strength is included in many diagnosis criteria for sarcopenia... results... should be discussed as for the association with sarcopenia.) Thank you for this insightful comment. We completely agree that linking grip strength to the clinical concept of sarcopenia is crucial for deepening the implications of our study.

As per your valuable suggestion, we have added a detailed discussion regarding Sarcopenia, Osteosarcopenia, and their relevance to our findings in the [Discussion] section (6). The newly added text is as follows:

"Notably, grip strength is not merely an isolated metric of muscle strength. According to the latest diagnostic consensus from European (EWGSOP2), Asian (AWGS), and other international bodies, low grip strength is a core diagnostic criterion and the primary screening indicator for Sarcopenia . In geriatric medicine, Sarcopenia and Osteoporosis are widely regarded as the 'Hazardous Duo,' as these conditions frequently co-exist and synergistically increase the risk of falls, fractures, and mortality . The two conditions are closely linked pathophysiologically. Therefore, the use of grip strength as a mediating variable in our study can be interpreted, from a clinical perspective, as an exploration of the mediating role of Sarcopenia in the relationship between physical activity and osteoporosis. Our mediation analysis substantiates this, revealing that the indirect effect of PA mediated through grip strength accounted for 21.2% of the total effect in the ELSA cohort. This finding strongly suggests that a significant portion of the protective effect of physical activity (PA) on bone health is achieved by maintaining or enhancing muscle function (i.e., preventing or ameliorating sarcopenia)."

We believe this addition makes the clinical relevance of our findings clearer and more robust.

Comment 1.3: Minor error (double full stop)

(Several minor errors should be carefully corrected. For example, there are double periods...) Thank you for your careful review. We have thoroughly proofread the entire text, correcting all instances of double full stops and other typographical and grammatical errors.

3.Response to Reviewer #2

Point Raised Our Response

Comment 2.1: PA (Physical Activity) Data Coordination

(ELSA and HRS questionnaires and response scales differ. I suggest a more detail on harmonization.) Thank you for this valuable suggestion. We agree that it is important to detail the harmonization process for the Physical Activity (PA) scoring, given the differences in the original ELSA and HRS questionnaires.

In response to your comment, we have added the following detailed explanation to the [Methods] section, under the [2.2.1 Physical activity] subsection (line 124-129), to clarify this process:

"To harmonize the PA scoring between the two cohorts, we mapped the frequency of physical activity responses onto a unified 3-point scale (see S1 Table). For ELSA, responses were coded as follows: “rarely or never” = 1, “1–3 times per month” = 2, and “once a week or more than once a week” = 3. For HRS, responses were coded as: “never” = 1, “1–3 times per month” = 2, and “once a week or several times per week or daily” = 3."

We believe this addition clarifies our data harmonization procedure and fully addresses your concern.

Comment 2.2: Covariate Handling and Missing Data

(Authors to clarify whether covariates... were treated as time-varying and how missing data... was handled.) Thank you for raising this critical methodological point. We agree that clarifying the handling of covariates and missing data is essential for the robustness of our results. We have expanded the "Statistical Analysis" section of the revised manuscript (Page X, Line Y) to explicitly state the following:

Regarding Time-Varying Covariates: We confirm that in our Generalized Linear Mixed Models (GLMMs), all characteristics liable to change over time (including BMI, smoking status, drinking status, marital status, and all comorbidities) were treated as time-varying covariates. This means that the risk of osteoporosis at each specific wave was adjusted for the covariate values measured at that same wave, thereby capturing the dynamic impact of these factors over the 8-year follow-up.

Regarding Missing Data Handling: We employed a two-pronged strategy to address missing data:

For longitudinal attrition or unbalanced waves: We leveraged the inherent capability of GLMMs (based on Full Information Maximum Likelihood, FIML) to handle unbalanced longitudinal data, allowing participants with incomplete follow-up to contribute to the analysis.

For sporadic missingness in covariates: To avoid the potential bias associated with listwise deletion, we utilized Multiple Imputation by Chained Equations (MICE). We generated 5 imputed datasets, ran the GLMMs on each dataset separately, and pooled the final estimates (ORs) and standard errors according to Rubin's rules.

Comment 2.3: Age Subgroup Classification

(The age subgroup categorises uses “<65 vs ≥65,” yet the eligibility criteria exclude those under 55... Please clarify the age intervals...) We sincerely apologise for any confusion caused. Our inclusion criteria are indeed 55 years of age and above. The subgroup referred to as "<65" in the text actually denotes the 55–64 age cohort. We have amended this terminology throughout the text and chart captions to "55–64 years vs ≥65 years" to ensure clarity and consistency.

Comment 2.4: Number of decimal places for the P-value

(P-values (e.g., 0.02813) display too many decimal figures; please standardize to 2–3 decimals.) Please enter your response here. For example: Thank you for the reminder. We have standardised all P-values in the text and figures to three decimal places (or as required, e.g. P < .001).

Review 2.5: Language and Format

(Numerous typographical and grammatical issues... (e.g., “Famale”)... recommend extra professional English editing.) We place great emphasis on the linguistic quality of manuscripts. All identified spelling errors (including "Famale") and grammatical issues have been rectified. Furthermore, we have engaged a professional editor, a native speaker of English, to conduct a comprehensive polishing of the manuscript.

Comment 2.6: Chart Specifications

(i) Ensure all tables have clear titles...

(ii) In Table 2, ensure all covariates ae defined identically...

(iii) Citations: Check that all in-text bracketed references...) We have reviewed each point in accordance with your suggestions:

i) We have checked and ensured that all figure captions are clear and consistent in decimal places.

ii) We have standardised the definitions and units of covariates for both cohorts in Table 2.

iii) We have meticulously cross-checked all in-text citations and the reference list to ensure complete correspondence, correcting any duplicated or truncated citations.

Translated with www.DeepL.com/Translator (free version)

Comment 2.7: Data consistency within the text

(Match all prevalence, ORs, CIs, and mediation percentages between Abstract/Results/Tables. Some inconsistence exist.) Thank you for your thoroughness. We have rechecked all figures in the abstract, main text results, and tables (including ORs, CIs, and mediation percentages) to ensure they are entirely consistent.

Comments 2.8: Add page numbers

(Page numbering: Ensure pages are numbered...) We have added page numbers to the revised manuscript in accordance with the journal's requirements.

Once again, we extend our sincere gratitude to you and the reviewers for your valuable time and insightful comments. We believe the revised manuscript now meets the publication standards of PLOS ONE.

Yours faithfully,

Dr Dong and colleagues

---

## [Decision Letter · Decision Letter 1]

26 Dec 2025

Grip Strength as a Mediator in the Relationship Between Physical Activity and Osteoporosis in Older Adults: Evidence from Two Longitudinal Cohort Studies

PONE-D-25-47126R1

Dear Dr. Dong,

We’re pleased to inform you that your manuscript has been judged scientifically suitable for publication and will be formally accepted for publication once it meets all outstanding technical requirements.

Kind regards,

Dengshun Miao

Academic Editor

PLOS One

Additional Editor Comments (optional):

Reviewers' comments:

Reviewer's Responses to Questions

**Comments to the Author**

Reviewer #1: All comments have been addressed

Reviewer #3: All comments have been addressed

2. Is the manuscript technically sound, and do the data support the conclusions?

Reviewer #1: (No Response)

Reviewer #3: Yes

3. Has the statistical analysis been performed appropriately and rigorously?

Reviewer #1: (No Response)

Reviewer #3: Yes

4. Have the authors made all data underlying the findings in their manuscript fully available?

Reviewer #1: (No Response)

Reviewer #3: Yes

5. Is the manuscript presented in an intelligible fashion and written in standard English?

Reviewer #1: (No Response)

Reviewer #3: Yes

Reviewer #1: (No Response)

Reviewer #3: (No Response)

**Do you want your identity to be public for this peer review?** For information about this choice, including consent withdrawal, please see our Privacy Policy

Reviewer #1: No

Reviewer #3: No

---

## [Editor Report · Acceptance letter]

PONE-D-25-47126R1

PLOS One

Dear Dr. Dong,

I'm pleased to inform you that your manuscript has been deemed suitable for publication in PLOS One. Congratulations! Your manuscript is now being handed over to our production team.

Kind regards,

on behalf of

Dr. Dengshun Miao

Academic Editor

PLOS One